# Pre-meiotic 21-nucleotide reproductive phasiRNAs emerged in seed plants and diversified in flowering plants

Suresh Pokhrel [1,2], Kun Huang[3], Sébastien Bélanger[1], Junpeng Zhan [1,4], Jeffrey L. Caplan[3], Elena M. Kramer[5] & Blake C. Meyers [1,2]✉

Plant small RNAs are important regulatory elements that fine-tune gene expression and maintain genome integrity by silencing transposons. Reproductive organs of monocots produce abundant phased, small interfering RNAs (phasiRNAs). The 21-nt reproductive phasiRNAs triggered by miR2118 are highly enriched in pre-meiotic anthers, and have been found in multiple eudicot species, in contrast with prior reports of monocot specificity. The 24-nt reproductive phasiRNAs are triggered by miR2275, and are highly enriched during meiosis in many angiosperms. Here, we report the widespread presence of the 21-nt reproductive phasiRNA pathway in eudicots including canonical and non-canonical microRNA (miRNA) triggers of this pathway. In eudicots, these 21-nt phasiRNAs are enriched in pre-meiotic stages, a spatiotemporal distribution consistent with that of monocots and suggesting a role in anther development. Although this pathway is apparently absent in well-studied eudicot families including the Brassicaceae, Solanaceae and Fabaceae, our work in eudicots supports an earlier singular finding in spruce, a gymnosperm, indicating that the pathway of 21-nt reproductive phasiRNAs emerged in seed plants and was lost in some lineages.

[1] Donald Danforth Plant Science Center, Saint Louis, MO, USA. [2] Division of Plant Sciences, University of Missouri-Columbia, Columbia, MO, USA. [3] Bio-Imaging Center, Delaware Biotechnology Institute, University of Delaware, Newark, DE, USA. [4] Department of Biology and Institute of Plant and Food Science, Southern University of Science and Technology, Shenzhen, Guangdong, China. [5] Department of Organismic and Evolutionary Biology, Harvard University, Cambridge, MA, USA. ✉email: bmeyers@danforthcenter.org

PhasiRNAs are generally produced by the action of 22 nucleotide (nt) miRNA triggers on polyadenylated messenger RNA (mRNA) or long noncoding RNA targets generated by RNA Polymerase II[1]. These target transcripts are subsequently processed by the action of RNA-DEPENDENT RNA POLYMERASE 6 (RDR6) and DICER-LIKE (DCL) proteins into duplexes of 21- or 24-nt small interfering RNAs (siRNAs). These siRNAs are "in phase" with one another; ï.e., the siRNAs map to the genome with regular spacing, the result of processive cleavage by Dicer from a long precursor (a *PHAS* precursor). The best characterized phasiRNAs in plants are *trans*-acting siRNAs, which function in development by regulation of auxin signaling[1].

Reproductive tissues of grasses contain both 21 and 24 nt phasiRNAs, derived from noncoding precursors encoded at hundreds of genomic loci. The 21 nt phasiRNAs are triggered by miR2118 and are mainly enriched in pre-meiotic anther tissues, during the stages at which the specification of cell fate occurs; they originate in the epidermal layer of the anther but accumulate subepidermally[2]. The 24 nt phasiRNAs are abundant in the meiotic stages of anther tissues and are largely triggered by miR2275, although in some monocots the miR2275 trigger is absent[3]. The spatiotemporal pattern of 21 nt phasiRNAs in male reproductive tissues has been well-described in maize and rice[2,4]. These reproductive, 21 nt phasiRNAs lack obvious or, at least, validated targets but play a role in photoperiod-sensitive male sterility in rice[5]. A mutant in rice of an Argonaute (AGO) protein that selectively binds these 21 nt phasiRNAs[6] is male sterile. These data demonstrate that the 21 nt reproductive pre-meiotic phasiRNAs and/or their functions are required for male fertility, and thus they are hypothesized to function in some important aspect of reproductive development.

The evolutionary origins of 21 nt reproductive phasiRNAs are poorly examined. Their presence in monocots is clear, as the 21 nt pre-meiotic phasiRNAs were first reported in grasses[7] and later described from just three loci in asparagus but still enriched in the middle layer, tapetum, and archesporial cells[3]. Despite extensive analyses, 21 nt reproductive phasiRNAs have not been reported in several well-studied plant families, including the Brassicaceae, Fabaceae, and Solanaceae, leading to our own prior assumption that they are absent from the eudicots. In earlier work from our lab, we described a set of 21 noncoding loci in Norway spruce that are targets of miR2118, produce 21 nt phasiRNAs, and are enriched in abundance in male cones[8]. This observation, as-yet unsupported or reproduced from analyses of non-monocot angiosperms, indicated the emergence of the 21 nt reproductive phasiRNAs outside of the monocots, in gymnosperms.

Here we describe the discovery of the pathway of pre-meiotic, 21 nt reproductive phasiRNAs in several eudicots as follows: wild strawberry (*Fragaria vesca*), rose (*Rosa chinensis*), the basal eudicot columbine (*Aquilegia coerulea*), and flax (*Linum usitatissimum*). We conclude that similar to the 24 nt reproductive phasiRNAs[9], the 21 nt reproductive pathway is widespread in angiosperms and may even, with origins in seed plants, have emerged prior to the 24 nt reproductive phasiRNAs.

## Results

### Pre-meiotic anthers of wild strawberry produce abundant 21 nt phasiRNAs.

We analyzed small RNAs (sRNAs) in tissues of wild (diploid) strawberry (*F. vesca*), as this plant was highly informative for our earlier analyses of 24 nt reproductive phasiRNAs[9]; in fact, we were considering it as a possible model system for the study of 24 nt phasiRNAs. We characterized 21 nt phasiRNAs in both vegetative and reproductive tissues of wild strawberry, and we identified 25 loci that give rise to 21 nt phasiRNAs ("21-

*PHAS*" loci) that were abundant specifically at the anther stage 7 (Fig. 1a). This stage corresponds to the pre-meiotic stage of anther development[10]. These unannotated loci were mostly predicted to be noncoding (Supplementary Data 1); they contained only one conserved sequence motif, which is the target site of miR11308-5p (henceforth, miR11308) (Fig. 1b, c). miR11308 has three mature variants (Fig. 1b, d), two of which derive in the genome from polycistronic precursors (Supplementary Fig. 1A, B and Supplementary Data 2). Mature miR11308 accumulation peaked at the anther stage 7 (Fig. 1b), similar to the peak of the reproductive-enriched 21 nt phasiRNAs. For these loci, the most abundant, phased 21 nt phasiRNA is generated from the cleavage site of miR11308 (or in a register spaced by 21 bp), as exemplified in Fig. 1e. These loci were distributed across all but chromosomes 2 and 3 of the *F. vesca* genome, whereas miR11308 originated from chromosome 6 (Supplementary Fig. 1C). We were surprised to find pre-meiotic 21 nt reproductive phasiRNAs, as we could find no record of a previous report of their presence in a eudicot.

In maize and rice, mature variants of the miR2118 family trigger production of the 21 nt reproductive phasiRNAs[2,4]. In many other species, the miR482 family, a predecessor and relative of the miR2118 family, triggers phasiRNAs from disease resistance genes[11]. However, in wild strawberry, only six miR2118/miR482-derived noncoding 21-*PHAS* loci are enriched during anther stages (Supplementary Data 1). We found 76 21-*PHAS* loci triggered by miR2118/miR482 in total; 65 loci are abundant in vegetative tissues, whereas 11 loci are enriched in reproductive tissues. These loci are mostly from protein-coding genes and are mainly similar to genes encoding disease resistance proteins, mirroring the 21-*PHAS* loci of other eudicots, such as soybean and Medicago[12,13] (Supplementary Fig. 2A and Supplementary Data 1). In strawberry, we found four precursors give rise to 15 mature variants of miR2118/miR482; these variants accumulate in all tissue types, with greater abundances found in seeds (Supplementary Fig. 2B, C). Therefore, unlike grasses, in wild strawberry, the majority of 21 nt reproductive phasiRNAs are triggered not by miR2118 but rather by a lineage-specific miRNA, miR11308. Furthermore, we validated the cleavage directed by miR11308 and miR482/2118 family members of these 21-*PHAS* loci using the degradome sequencing method known as nanoPARE[14] (Supplementary Fig. 3A, B and Supplementary Data 3).

We next asked where this trigger of 21 nt reproductive phasiRNAs localizes. In situ hybridization localization of miR11308 in wild strawberry showed that it localizes to microspore mother cells (MMCs), meiocytes, and tapetal cells (Fig. 2a, b) unlike the miR2118 in maize, which localizes only in the epidermis[2]. miR11308 is more abundant in MMCs of pre-meiotic cells and in tapetal cells of meiotic stages compared to the post-meiotic stage. Next, we examined the localization patterns of the most abundant 21 nt phasiRNAs from these loci with single molecule fluorescence in situ hybridization (smFISH) (Supplementary Fig. 4A) using a pool of fluorescently labeled probes against 50 21 nt phasiRNAs. Surprisingly, we found these molecules were localized in all cell layers, most abundant in the MMC during the pre-meiotic stage of anther development (Fig. 3a). By the tetrad stage, the phasiRNAs are only detectable in the MMCs and barely detectable in other cell layers. This spatiotemporal distribution of 21 nt phasiRNAs is generally similar to the accumulation pattern found in grasses[2,15] and thus these 21 nt phasiRNAs in strawberry may play a role in male fertility.

### Conservation of miR11308 and 21 nt reproductive phasiRNAs in other Rosaceae species.

To determine whether this pathway is conserved in other members of the Rosaceae family and beyond,

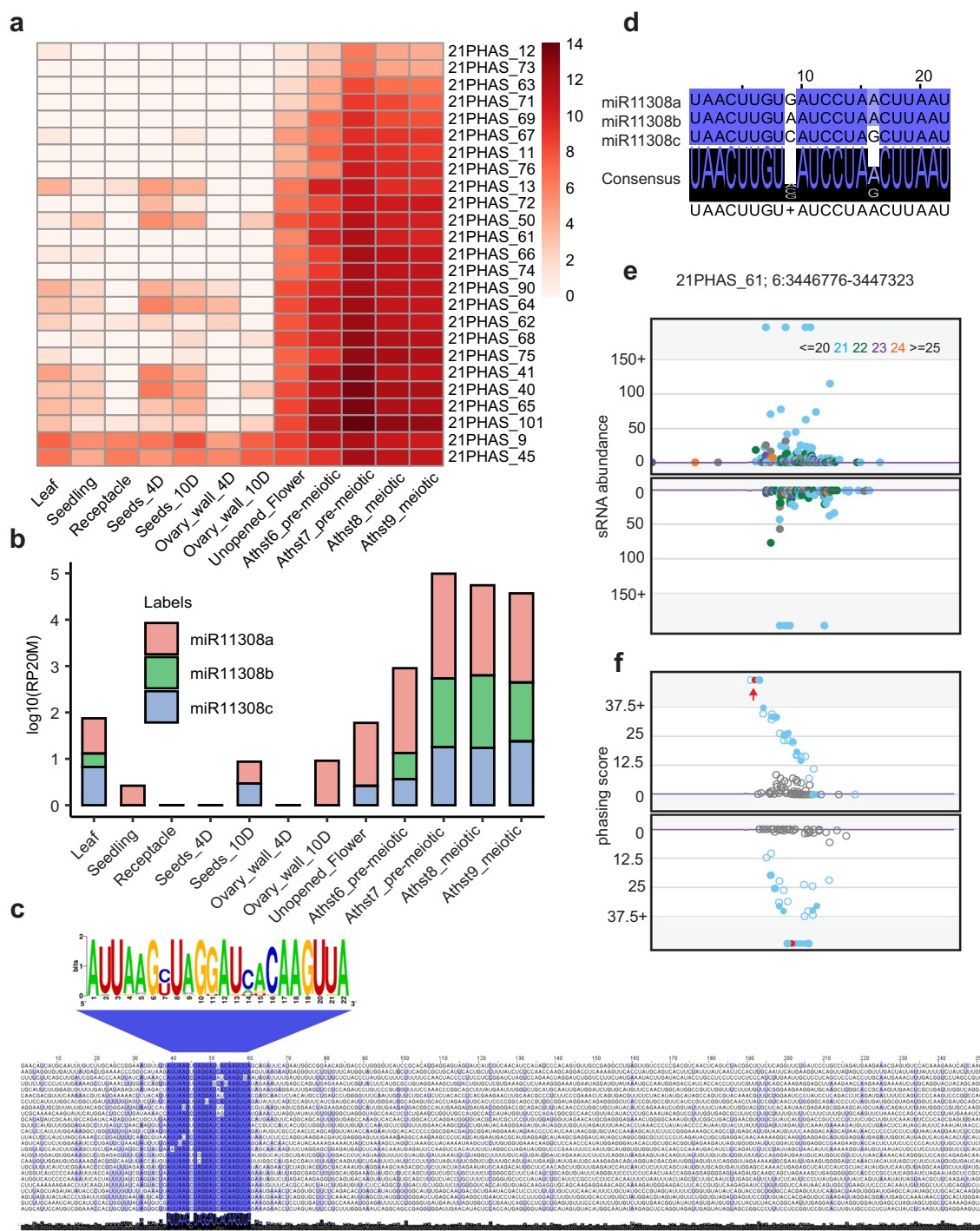

**Fig. 1 Reproductive 21 nt phasiRNAs triggered by miRNA miR11308 are abundant in wild strawberry. a** Expression of 21 nt reproductive phasiRNAs in different tissues and anther development stages. The key at the right indicates the abundance in units of log2(RP20M). RP20M: reads per 20 million mapped reads. Athst: anther stage. **b** Abundance of miR11308 members in log10(RP20M) in different tissues of wild strawberry. **c** Above: sequence logo denoting conservation of target site of miR11308 for 25 21-*PHAS* loci. Below: nucleotide sequence alignment of 21-*PHAS* loci with sequence similarity denoted by the intensity of the blue color showing that the miR11308 target site is the only conserved region for all the precursors. **d** Alignment of members of the miR11308 family in wild strawberry. The degree of conservation is denoted by intensity of the blue color; the consensus sequence of the alignment is shown with a sequence logo. **e** Abundance (RP15M) of small RNAs in both strands of an example locus padded with 500 base pairs, each side. sRNA sizes are denoted by colors, as indicated at the top. **f** Phasing score of same locus as **e**; the red dot indicates the highest phased sRNA position. The red dot also represents the coordinate (3,446,776), which exactly coincides with the predicted cleavage site of miR11308, denoted by the red arrow. Source data underlying Fig. 1a, b are provided as a Source Data file.

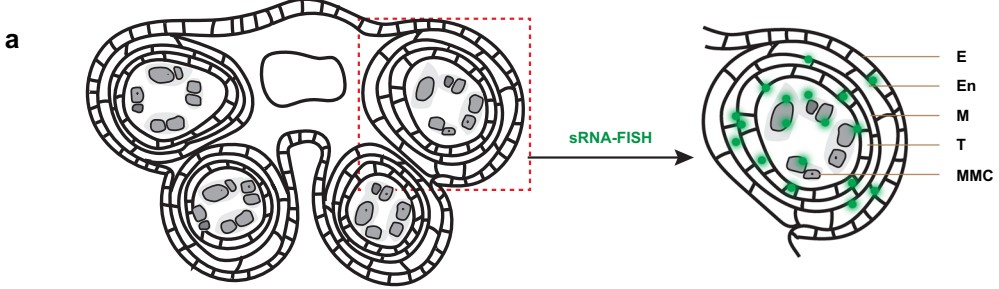

**Fig. 2 miR11308 accumulates in microspore mother cells, meiocytes, and tapetal cells in the anthers of wild strawberry. a** Schematic diagram of the sRNA-FISH method in anther stages shown in **b**. E: epidermis, En: endodermis, M: middle layer, T: tapetum, MMC: microspore mother cell. **b** In situ hybridization of miR11308 in pre-meiotic, meiotic, and post-meiotic anther stages. The experiment was repeated twice with similar results.

we investigated the presence of the miR11308 in species of Rosaceae and Betulaceae for which a genome sequence is available. We found that the pattern of miR11308 presence is consistent with its emergence in the Rosaceae subfamily Rosoideae, as it is present in at least four genera (Fig. 3b and Supplementary Data 2). Similar to miR2275[9], this miRNA generates mature sequences with a polycistronic precursor cluster, which itself is conserved in the Rosoideae. We hypothesize that this polycistronic nature of the precursors is of functional importance for the biogenesis of reproductive phasiRNAs.

We next analyzed the sRNA transcriptomes of rose and identified nine 21-*PHAS* loci enriched in the anthers of early reproductive bud stages (Fig. 3c and Supplementary Fig. 4B), thus confirming the presence of 21 nt reproductive phasiRNAs in this species. Similar to wild strawberry, these loci are mostly noncoding (Supplementary Data 4) and are also targeted by two variants of miR11308 (Fig. 3d). We further found that three precursors of six mature variants of the rose miR2118/482 family target 86 21-*PHAS* loci; we found them to be abundant in all tissue types and mostly derive from genes coding for nucleotide-

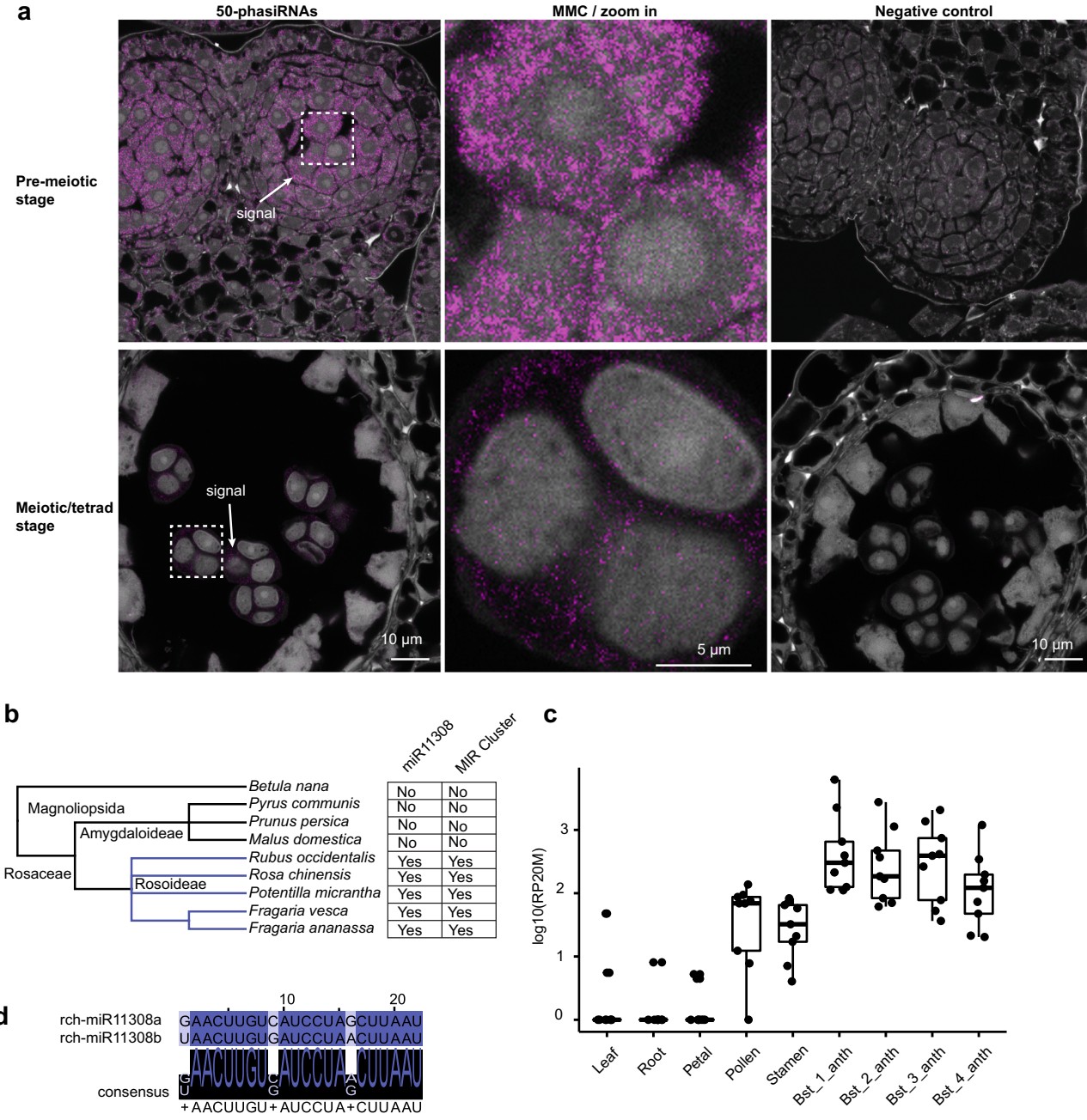

**Fig. 3 The 21 nt reproductive phasiRNAs localize in all cell layers of the pre-meiotic strawberry anthers and the miR11308 trigger emerged in the Rosoideae. a** Localization of abundant 50 21 nt reproductive phasiRNAs in pre-meiotic and meiotic anther tissues of wild strawberry. The localization experiment was repeated twice with similar results. **b** Phylogenetic tree showing conservation of miR11308 in the Rosaceae. **c** Abundances of summed 21 nt phasiRNAs in different tissues of rose in units of log10(RP20M). In the boxplot, the center line represents the median, box limits are the upper and lower quartiles; whiskers are the 1.5× interquartile range of upper or lower quartiles; points show the scatter of data points for nine ($n = 9$) 21-*PHAS* loci. Bst indicates bud stage, anth indicates anther. **d** Alignment of members of miR11308 in rose. Source data underlying Fig. 3c are provided as a Source Data file.

binding leucine-rich repeat proteins otherwise known as "NLRs," which commonly function as innate immune receptors (Supplementary Fig. 4C, D and Supplementary Data 4). Hence, we report that the miR11308 has acquired a role to generate 21 nt reproductive phasiRNAs in the Rosaceae subfamily Rosoideae.

**Trigger miRNAs and conservation of 21 nt reproductive phasiRNAs in the basal eudicot columbine and in flax.** We next examined the sRNA transcriptomes of vegetative tissues and four different bud stages (Supplementary Fig. 5A) of the basal eudicot *Aquilegia* (common name "columbine") to determine whether 21

nt reproductive phasiRNAs are present and/or conserved in this species. We identified 112 21-*PHAS* loci; 91 enriched in reproductive tissues (Fig. 4a) and an additional 21 21-*PHAS* loci that were expressed similarly in all tissue types (Supplementary Fig. 5B). Out of 112 loci, we found that 65 loci are targeted by miR14051; most loci (63/65) are from noncoding regions of the genome (Supplementary Data 5) and they possess a conserved 22 nt motif: the target site of miR14051 (Supplementary Fig. 5C). The trigger-target cleavage was detected in 33 (of 65) 21-*PHAS* loci using nanoPARE sequencing (Supplementary Data 6). The miR14051 miRNA has six mature variants from seven different

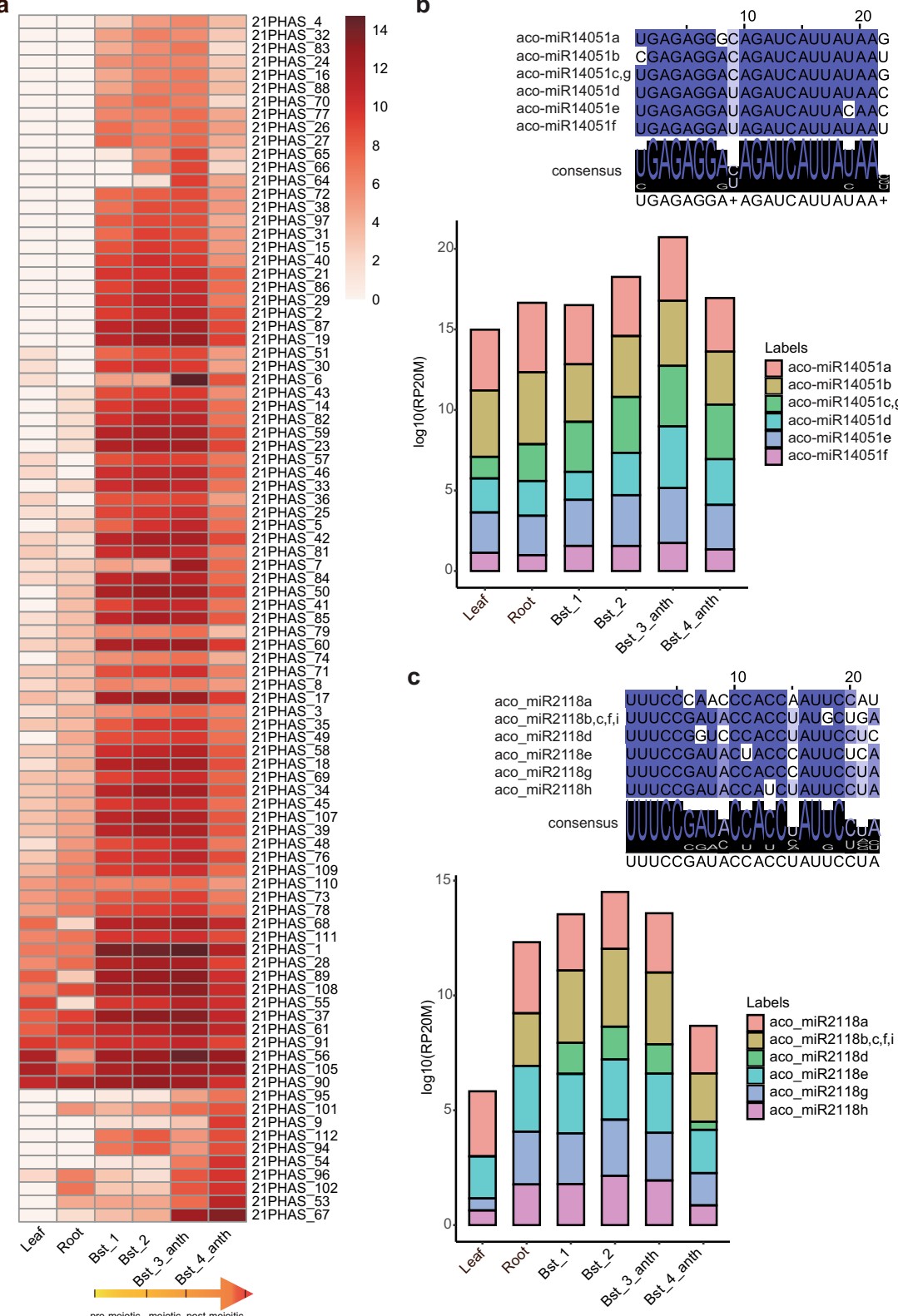

**Fig. 4 Conservation of reproductive 21 nt phasiRNAs and their triggers in the basal eudicot columbine. a** Expression of 21 nt reproductive phasiRNAs in different vegetative tissues and anther development stages in columbine. The lowermost ten loci are most enriched in stage 4. The yellow gradient arrow indicates the developmental stages of anthers. The key at the right indicates the abundance in units of log2(RP20M). Bst indicates bud stage, anth indicates anther. **b** Above: alignment of members of miR14051 in columbine. The degree of conservation is denoted by the intensity of the blue color and the consensus sequence of the alignment is shown in a sequence logo. Below: abundance of miR14051 members in log10(RP20M) in different tissues of columbine. **c** Similar to **b** but for the miR2118/482 family in columbine. Source data are provided as a Source Data file.

precursors (Fig. 4b and Supplementary Data 2), with one poly-cistronic cluster generating two mature variants (Supplementary Fig. 5D). The accumulation pattern of miR14051 is similar to the 21 nt phasiRNAs initiated by this miRNA (Fig. 4a, b).

The remaining 47 (of 112) 21-*PHAS* loci are targeted by mature variants of the miR2118/482 family (Fig. 4a, c and Supplementary Fig. 6A). Among these, 37 loci are abundant in bud stage 2 and 3 (Fig. 4a), and are noncoding, whereas the other 10 loci are abundant in all tissue type and are mostly derived from NLR protein-coding genes (Supplementary Fig. 5A and Supplementary Data 5). NanoPARE sequencing was used to validate the trigger-target interaction of 25 (of 47) 21-*PHAS* loci (Supplementary Data 6). The miR2118/482 family has six mature variants that are derived from nine different precursors with two polycistronic clusters (Fig. 4c, Supplementary Fig. 6B, and Supplementary Data 2). The miR2118/482 family members are highly abundant at bud stages 2 and 3, similar to the phasiRNAs they trigger (Fig. 4c). Most of the reproductive 21-*PHAS* loci in *Aquilegia* are concentrated on chromosome 4, an unusual chromosome showing higher polymorphism compared to other chromosomes between the ten different *Aquilegia* species[16]. The miR14051 miRNA precursors were derived from chromosomes 4 and 1, whereas miR2118/482 precursors were distributed on chromosomes 1, 3, and 5 (Supplementary Fig. 7). Unlike grasses, in *Aquilegia*, 21 nt reproductive phasiRNAs are triggered by two miRNA family members, for which at least one precursor is polycistronic.

To more precisely identify the developmental stages of the buds/anthers that we sequenced, anthers from seven bud lengths were examined. We found anthers from 2 to 4 mm buds are in pre-meiotic stages, whereas 5–6 and 10–20 mm buds are meiotic and post-meiotic stages, respectively (Supplementary Fig. 8). Therefore, the 21 nt phasiRNAs in columbine initiate in pre-meiotic stages and their accumulation is maintained up to post-meiotic stages.

To further confirm the conservation of the 21 nt reproductive phasiRNA pathway in eudicots, we profiled sRNAs from both vegetative and reproductive tissues of flax, a species in the phylogenetic tree of eudicots outside the Rosaceae but within the rosids. We found 11 reproductive-enriched 21-*PHAS* noncoding loci triggered by miR2118/482 variants (Supplementary Fig. 9A–C) in flax (Supplementary Data 2 and 7). Unlike columbine, in flax, these 21-*PHAS* loci are triggered only by the canonical trigger of pre-meiotic phasiRNAs, as described from extensive work in monocots[2,4] (miR2118/miR482).

**Genes important to the 21 nt reproductive phasiRNA pathway.** Factors known to function in phasiRNA biogenesis are mainly RDR6, AGOs, and DCL proteins. We analyzed the presence of these proteins in a total of 13 species: 4 eudicots (wild strawberry, rose, columbine, flax) we analyzed for phasiRNAs, 2 gymnosperms (Norway spruce, ginko), and another 7 representative species for which these genes were already characterized—*Amborella*, soybean, tomato, *Arabidopsis*, *Asparagus*, maize, and rice (Fig. 5a and Supplementary Fig. 10A, B). RDR6 is responsible for making the double-stranded RNA precursors after miRNA cleavage during phasiRNA biogenesis and it is conserved in seed plants (Supplementary Fig. 10A). Among 13 species, RDR6 is present in two or more copies in all species, except in rice, *Asparagus*, *Arabidopsis*, and columbine. The three eudicot species we analyzed for phasiRNAs have two *RDR6* copies, but not columbine (Supplementary Fig. 10A). We hypothesize that there is a species-specific duplication of RDR6, potentially facilitating a sub-functionalization of their roles, as both copies are enriched in reproductive tissues (Supplementary Data 8). The 21 nt

phasiRNAs are produced by *DCL4*, which is conserved in seed plants (Supplementary Fig. 10B) and mostly enriched in reproductive tissues, except in rose (Supplementary Data 8). DCL3 and DCL5 are Dicer proteins responsible for the production of 24 nt siRNAs, with DCL5 specialized for 24 nt phasiRNAs in monocots and DCL3 hypothesized to functioning in dual roles, producing 24 nt phasiRNAs in eudicots in addition to its well-described role in making heterochromatic siRNAs[2,9]. There is a duplication of *DCL3* in all four eudicot genomes that we examined for pha-siRNAs and in a gymnosperm, ginko, among all the species; duplicated *DCL3* (*DCL3b*) was more enriched in reproductive tissues than *DCL3a* (Supplementary Data 8), suggesting the possibility of neo-functionalization of DCL3b for the production of 24 nt reproductive phasiRNAs in eudicots[9].

We identified 12, 13, and 17 AGO proteins encoded in the wild strawberry, rose, columbine, and flax genomes, respectively (Fig. 5a). All of the identified *AGO* transcripts are expressed ≥0.5 TPM (transcripts per million) in these four species (Fig. 5b). In rice, AGO1b/1d were implicated in loading miR2118 and U-21-nt phasiRNAs[15], whereas AGO5c (i.e., rice MEL1) is implicated for the loading of reproductive C-21-nt phasiRNAs[6,15]. Here we identified two *AGO5* copies in strawberry and both *AGO5* copies are enriched in pre-meiotic stages, whereas in rose, among three *AGO5* copies, only *AGO5b* (orthologous to *AGO5b* in strawberry) was enriched in reproductive tissues. In flax, one copy of *AGO5* is enriched in the reproductive tissues. In columbine, both copies of *AGO5* are enriched in the reproductive tissues but *AGO5b* was more enriched (Supplementary Data 8). Overall, based on phylogeny and their expression profiles, AGO5 or AGO5a/b are the candidate AGO proteins for loading pre-meiotic 21 nt phasiRNAs in eudicots as they are in monocots. Among AGO1 family members, in columbine, *AGO1c* is more enriched in reproductive stages, whereas in flax, strawberry, and rose, *AGO1a* is more enriched in the reproductive tissues (Fig. 5b and Supplementary Data 8). Thus, AGO1c for columbine, and AGO1a for other three eudicots might be the effector protein for triggers of 21 nt reproductive phasiRNAs. Orthologs of AGO18 are absent in eudicots, gymnosperms, *Amborella*, and *Asparagus* consistent with its hypothesized origin in grasses (Fig. 5a). Genes encoding members of the AGO6 clade (specifically *AGO6a*) were enriched more than *AGO4* in the reproductive tissues, whereas *AGO9* in rose and strawberry are expressed exclusively in the reproductive tissues (Fig. 5b and Supplementary Data 8). It is known that AGO4, AGO6, or AGO9 are binding molecules for 24 nt heterochromatic siRNAs during RNA-directed DNA methylation[17]. Thus, based on their gene expression and enrichment, AGO9 and AGO6 may also act as the effector molecule for reproductive 24 nt phasiRNAs in eudicots.

## Discussion

We found that 21 nt reproductive phasiRNAs are present in several eudicots: wild strawberry, rose, the basal eudicot colum-bine, and flax. Whereas previous studies have shown that 21 nt reproductive phasiRNAs exist in one gymnosperm[8] and more widely in monocots[2,4,7,18,19] (Fig. 6a), this pathway has not been described in several well-studied eudicot families including the Brassicaceae, Fabaceae, and Solanaceae[2,12,13]. Another class of 21 or 22 nt secondary siRNAs active in plant reproduction are the epigenetically activated siRNAs (easiRNAs) that accumulate in *Arabidopsis* during stages of pollen maturation, derived from activated transposable elements[20,21]. easiRNAs appear to play a role in male gamete production and possibly post-fertilization genome stability and seed viability[22]. Even though both pha-siRNAs and easiRNAs are secondary siRNAs important for plant reproduction, there is no evidence suggesting that they are related

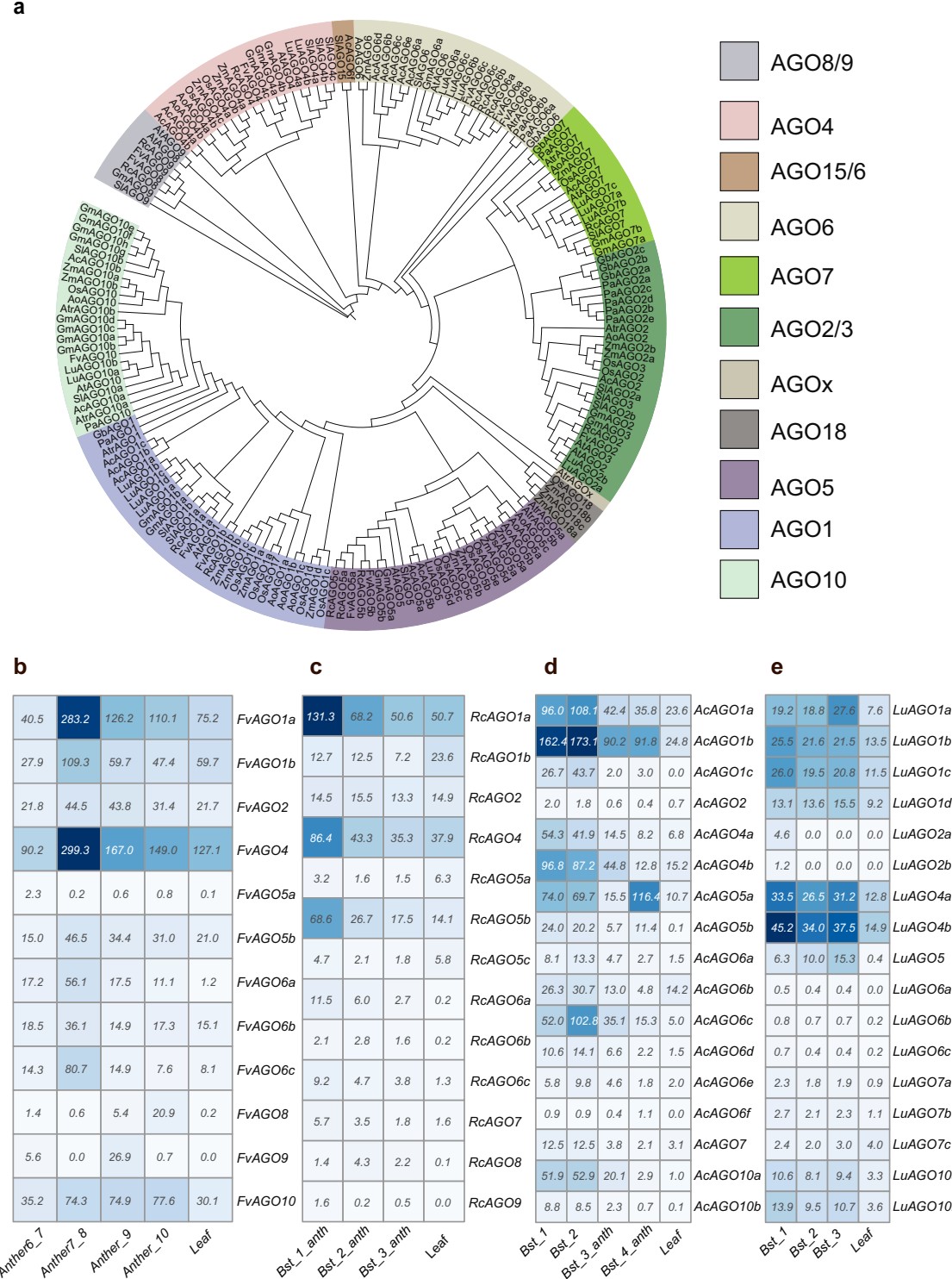

**Fig. 5 Characterization of AGO gene family members in wild strawberry, rose, columbine, flax, and other representative species from gymnosperms, eudicots, and monocots. a** Phylogenetic tree of Argonaute protein family members encoded in the genomes of plant species, with two-letter prefix indicating the source genome: wild strawberry (Fv), rose (Rc), columbine (Ac), flax (Lu), Norway spruce (Pa), and ginko (Gb) identified in this study along with other seven representative species—*Amborella* (Atr), soybean (Gm), tomato (Sl), *Arabidopsis* (At), asparagus (Ao), maize (Zm), and rice (Os). Abundance profile of genes encoding AGO family members in **b** wild strawberry, **c** rose, **d** in columbine, and **e** in flax. Bst indicates the bud stage, anth indicates the anther. Source data underlying Fig. 5b–e are provided as a Source Data file.

**a**

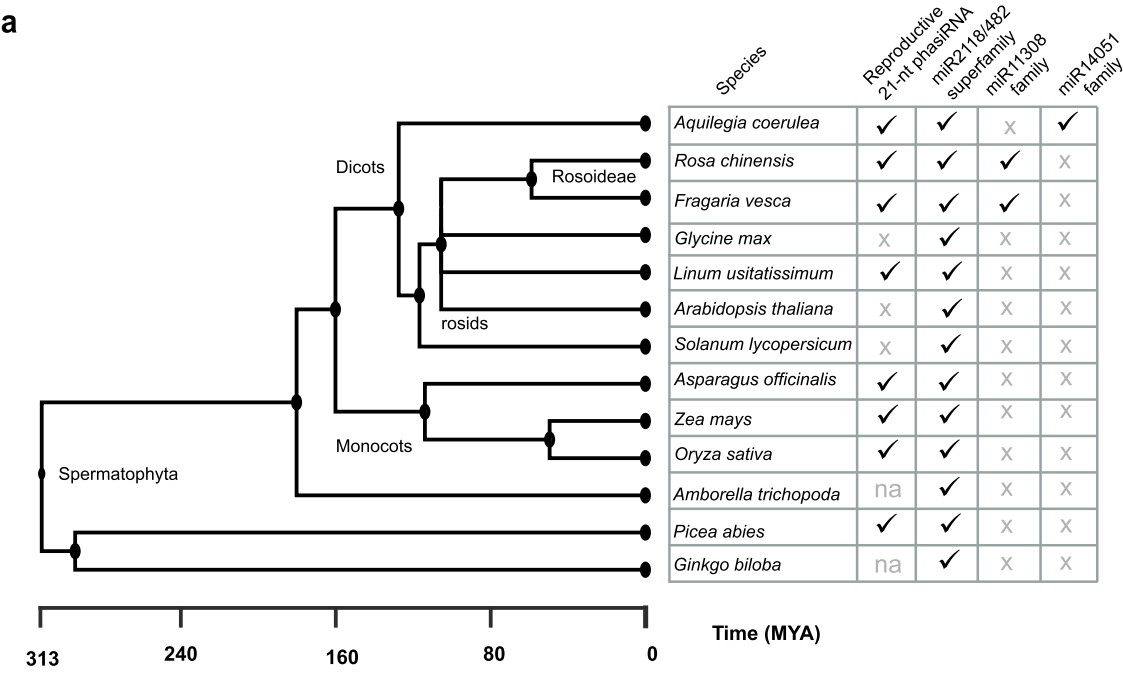

| Species | Reproductive 21-nt phasiRNA | miR2118/482 superfamily | miR11308 family | miR14051 family |
|---|---|---|---|---|
| *Aquilegia coerulea* | ✔ | ✔ | ✗ | ✔ |
| *Rosa chinensis* | ✔ | ✔ | ✔ | ✗ |
| *Fragaria vesca* | ✔ | ✔ | ✔ | ✗ |
| *Glycine max* | ✗ | ✔ | ✗ | ✗ |
| *Linum usitatissimum* | ✔ | ✔ | ✗ | ✗ |
| *Arabidopsis thaliana* | ✗ | ✔ | ✗ | ✗ |
| *Solanum lycopersicum* | ✗ | ✔ | ✗ | ✗ |
| *Asparagus officinalis* | ✔ | ✔ | ✗ | ✗ |
| *Zea mays* | ✔ | ✔ | ✗ | ✗ |
| *Oryza sativa* | ✔ | ✔ | ✗ | ✗ |
| *Amborella trichopoda* | na | ✔ | ✗ | ✗ |
| *Picea abies* | ✔ | ✔ | ✗ | ✗ |
| *Ginkgo biloba* | na | ✔ | ✗ | ✗ |

**b**

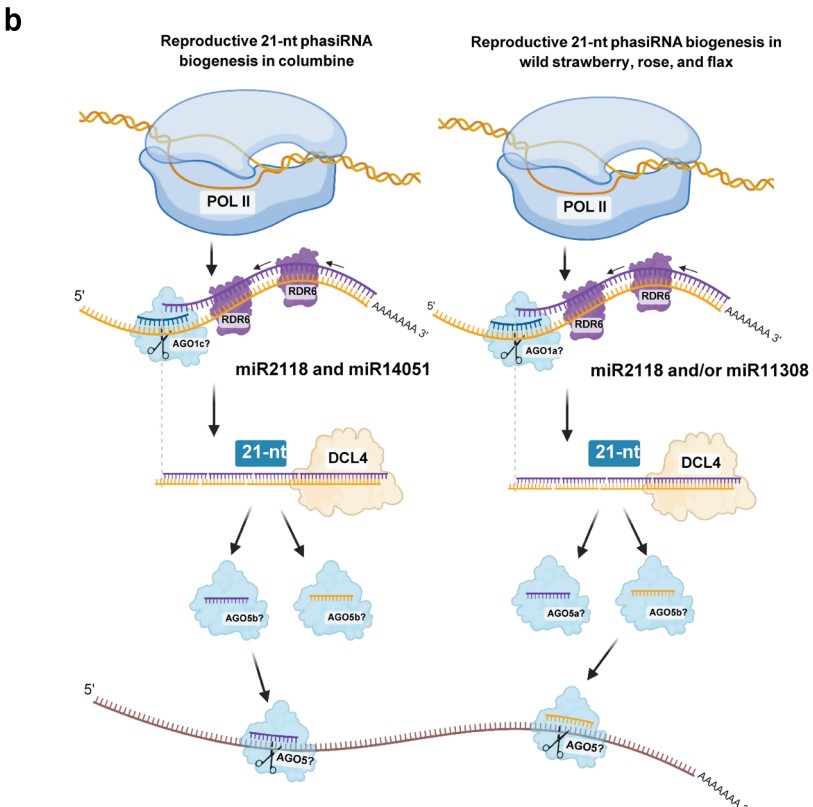

classes of RNAs, nor is it evident that easiRNAs are found outside of *Arabidopsis*. We hypothesize that 21 nt reproductive phasiRNAs are a more fundamental and broadly conserved pathway, which emerged in gymnosperms but were lost in some lineages due to accommodations or adaptations of development.

In the basal eudicot columbine, we showed that reproductive tissues produce abundant phasiRNAs generated by both the canonical trigger miR2118/482 and a lineage-specific miRNA, miR14051, indicating a division of this phasiRNA biogenesis role.

Moreover, the miR2118/482 family triggers 21 nt phasiRNAs from *NLR* genes in columbine, a conserved role of this miRNA from gymnosperms to angiosperms, particularly in eudicots[1]. The miR2118/482-derived *NLR* phasiRNA pathway is also found in wild strawberry and rose; although the role of this miRNA family in generating reproductive 21 nt phasiRNAs is almost completely shifted to miR11308, a non-canonical trigger of reproductive phasiRNAs, we also found evidence for the canonical trigger of reproductive phasiRNAs in wild strawberry miR2118. In flax, we

**Fig. 6 Conservation of reproductive 21 nt phasiRNA in seed plants. a** Phylogeny showing the representative species of seed plants according to estimated divergence times based on the time tree of life (Kumar et al.[51]). The miR2118/482 superfamily consists of miR2118, miR482, miR8558, and miR472. MYA means million years ago. √ indicates the presence and X indicates the absence; na indicates "not analyzed". **b** Model of generation of 21 nt reproductive phasiRNAs in seed and flowering plants. miRNA triggers initiate the cleavage of Pol II transcripts with the help of AGO proteins resulting 3′-end cleaved fragments, which are processed by RDR6 to generate double-stranded RNA molecules. DCL4 chops these double-stranded molecules into 21 nt duplexes, initiated from the cleavage site (denoted by dotted lines). In gymnosperms, monocots, and some eudicots such as flax, miR2118 is the only trigger for the production of 21 nt reproductive phasiRNAs, whereas in columbine, rose, and wild strawberry, there is a lineage-specific miRNA trigger for this role. Based on the roles in other species, family members of AGO1 and AGO5 might load the trigger miRNA and 21 nt phasiRNAs, respectively. The resulting phasiRNAs from these mechanisms were predicted to act in both *cis* and *trans* to modulate the gene expression based on the studies in monocots. Panel **b** was created with BioRender.com under an academic license.

found that only miR2118/482 triggers the production of 21 nt reproductive phasiRNAs, consistent with a conserved role of the canonical trigger in this species. Based on these data, we propose a model of how different triggers in different species function to generate reproductive 21 nt phasiRNAs in seed and flowering plants (Fig. 6b). All of these triggers starts with uracil (U) and are 22 nt in length consistent with earlier finding that only the 22 nt form of miRNAs in AGO1 complex are capable of initiating phasiRNA biogenesis[23]. Based on the roles in other species[6,24,25], family members of the AGO5 clade might load 21 nt phasiRNAs that were predicted to act in both *cis* and *trans*, to modulate transcript levels. Clearly, more extensive data, from more diverse genomes of seed plants, would help improve the granularity of this model.

Similar to the 21 nt reproductive phasiRNAs in grasses, in wild strawberry, these phasiRNAs are also localized in all cell layers and are abundant in MMCs of pre-meiotic anther, suggesting similar role in reproductive development of eudicot anthers— whatever that role may be. Even more than a decade after describing the 21 nt reproductive phasiRNAs in rice and maize[7], their precise molecular activities are unclear. Unlike miR2118 in grasses, which localizes to the epidermis, miR11308 localizes in MMCs in pre-meiotic anthers, suggesting MMCs as the origin of these pre-meiotic phasiRNAs. The localization of miR11308 in tapetal cells of meiotic anthers suggests other potential roles of this miRNA in later stages of anther development, yet to be examined.

In conclusion, the emergence and conservation of 21 nt reproductive phasiRNAs in eudicots supports an origin in the seed plants and suggests a functional role in reproduction. In rice, mutations in these 21-*PHAS* loci cause photoperiod-sensitive male sterility[5], a key feature that has been leveraged for hybrid seed production. Therefore, the study of these 21-*PHAS* loci in a broader set of plant species may open up avenues to improve plant yields. Currently, it is unknown how these phasiRNAs modulate male fertility; future studies are needed to produce a deeper understanding of reproductive phasiRNAs. Some species may be better models than others, due to their life history, miRNA triggers, *PHAS* locus complexity, or other traits, as demonstrated by the diversity we observed in eudicots.

## Methods

**Plant material harvesting and RNA isolation**. Vegetative tissues and unopened flower buds/anthers of columbine (cultivar, "origami") were collected from greenhouse-grown plants under conditions of 20/18 °C day/night and 16/8 h light/ dark. A total of four different bud stages (Supplementary Fig. 5A) were char- acterized and buds were harvested for stage 1 and 2, and from stage 3 and 4, anthers were harvested. Two biological replicates were sequenced for each stage. Flax was grown in a growth chamber with 12 h light at 25 °C followed by 12 h dark at 20 °C. Three bud stages (Supplementary Fig. 8A) were harvested and frozen immediately in liquid nitrogen. Rose anthers from four different bud stages (Supplementary Fig. 4B) were harvested and immediately frozen in liquid Nitrogen. Total RNA was isolated using the PureLink Plant RNA Reagent (ThermoFisher Scientific, catalog number #12322012) following the manufacturer's instructions. Total RNA quality and quantity were assessed by running formaldehyde-containing agarose gel and

using a Qubit Fluorometer (ThermoFisher Scientific, catalog number #Q33238). Flax sRNAs (20–30 nt) were size-selected in a 15% polyacrylamide/urea gel and used for sRNA library preparation, whereas for other species direct total RNA was used for sRNA library construction. An aliquot of 5 µg of total RNA was used for size selection.

**RNA libraries and sequencing**. Flax sRNA libraries were constructed using the NEBNext Small RNA Library Preparation set for Illumina (NEB, catalog number #E7300); for columbine and rose, RealSeq-AC miRNA Library kit for Illumina (Somagenics, catalog number #500-00012) was used as per the manufacturer's recommendation. RNA-sequencing (RNA-seq) libraries were constructed using NEBNext Ultra II Directional RNA Prep kit for Illumina; RNA was treated with DNase I (NEB, catalog number #M0303S) and then cleaned using the RNA Clean & Concentrator™-5 (Zymo Research, catalog number #R1015S). sRNA- and RNA- seq libraries were single-end sequenced with 76 cycles. All libraries were sequenced on an Illumina Nextseq 500 instrument at the University of Delaware Sequencing and Genotyping Center at the Delaware Biotechnology Institute. For strawberry vegetative/reproductive tissues[9,26,27] and rose vegetative tissues[28], sRNA- and RNA-seq data were downloaded from public databases and used for the analysis.

**sRNA data analysis and *PHAS* annotation**. Raw reads of sRNA libraries were processed by an in-house preprocessing pipeline[29]. PHASIS[30] with a *p*-value cutoff of 0.001 and ShortStack[31] with default parameters were used to identify the *PHAS* loci in all of the species. *PHAS* loci from two different software were anno- tated using BEDTools[32]. Only *PHAS* loci having target sites for miRNA triggers were anno- tated as valid *PHAS* loci. Target prediction was carried out by sPARTA[33] with a target cutoff score of ≤4. The sRNA abundance and phasing score were viewed at customized browser[34]. Based on the target site of triggers, 50 bp upstream and 500 bp downstream strand-specific sequences were extracted from each *PHAS* loci and were annotated de novo by BLASTX[35] against UniRef90. The coding potential of *PHAS* loci were assessed for protein-coding potential by CPC (Coding Potential Calculator)[36].

**RNA-seq data analysis and visualization**. Raw reads were processed by an in- house preprocessing pipeline and Hisat2[37] was used to align the reads with respective genomes using default parameters. Raw counts were obtained by using TPMCalculator[38] and normalized to TPM. To visualize the conservation of the target site of miRNA triggers, a multiple sequence alignment of *PHAS* loci strand- specific sequences was performed using MUSCLE[39] with default parameters and we visualized the alignment using Jalview[40]. Weblogo images were created using logo generation form (http://weblogo.berkeley.edu/logo.cgi). Circular plots were made using OmicCircos[41] for the chromosomal distributions, and pheatmap (https://rdrr.io/cran/pheatmap/) and ggpubr (https://github.com/kassambara/ ggpubr) were used to draw heatmaps and boxplot in R.

**Phylogenetic analysis**. Identification of RDR, Dicer, and AGO families for all of the species were carried out by using Orthofinder[42], except for Norway spruce (*Pica abies)* for which genBlastG[43] was used. The resulting protein sequences were visualized in CDvist[44] to verify complete domains. Curated proteins were aligned using default settings in PASTA[45]. The maximum likelihood gene tree for each protein family were generated using RAxML[46] over 100 rapid bootstraps with options "-x 12345 -f a -p 13423 -m PROTGAMMAAUTO." Trees were visualized and manipulated in iTOL[47]. The annotations used for the phylogenetic analysis for all 13 species for AGO, DCL, and RDR proteins are listed in Supplementary Data 9–11, respectively.

**Tissue embedding and microscopy**. Fresh anthers were dissected from different bud stages and fixed in a FAA (Formaldehyde Alcohol Acetic Acid, 10%:50%:5% + 35% water) solution overnight and dehydrated through a standard acetone series (30%, 50%, 70%, 80%, 90%, and 100% of cold acetone) prior to being resin infil- trated and embedded using the Quetol (Electron Microscopy Sciences, catalog number #14640) using either heat polymerization. Embedded tissues were sec- tioned at 0.5 µm using the Leica Ultracut UCT (Leica Microsystems, Inc.) and

stained using a 0.5% Toluidine Blue O dye (Electron Microscopy Sciences, catalog number #26074-15). Microscopy images were captured using a ZEISS Axio Zoom. V15 microscope using the PlanNeoFluar Z 2.3×/0.57 FWD 10.6 mm objective lens with a magnification of ×260. Digital images were captured at 2584 × 1936 pixel resolution at 12 bit/channel.

**sRNA fluorescent in situ hybridization**. The sRNA fluorescent in situ hybridization (sRNA-FISH) experiment was carried out as described before[48]. Briefly, fresh unopened buds of strawberry were dissected and fixed in a 20 ml glass vial using 4% paraformaldehyde in 1× PHEM buffer (5 mM HEPES, 60 mM PIPES, 10 mM EGTA, 2 mM $MgSO_4$ pH 7). Fixation was done in a vacuum chamber at 0.08 MPa for three times, 15 min each. After fixation, samples were sent for paraffin embedding at histology lab from Nemours/Alfred I. duPont Hospital for Children (Wilmington, DE). Samples were sectioned using a paraffin microtome and dried on poly-L-lysine-treated coverslips. FISH hybridization was modified from the protocol by Javelle et al.[49] by replacing the antibody with primary anti-Digoxigenin Fab fragment (Sigma-Aldrich catalog number #11214667001) and secondary donkey anti-sheep IgG (H + L) AF647, AF568, or AF633 (ThermoFisher Scientific catalog number #A-21448, A-21099, and A-21100); the dilution factors for the primary incubation were 1 : 100 in washing buffer and for secondary incubation were 1 : 1000. Briefly, samples were de-paraffinized using Histo-Clear (Fisher scientific, 50-899-90147) and re-hydrated by going through an ethanol series of 95%, 80%, 70%, 50%, 30%, and 10% (vol/vol) (30 s each) and through water (1 min) at room temperature. After protease (Sigma, P5147) digestion (20 min, 37 °C), samples were treated with 0.2% glycine (Sigma-Aldrich, catalog number #G8898) for 2 min. After two washes in 1× phosphate-buffered saline, samples were dehydrated and then hybridized with probes overnight at 53.3 °C. Ten milliliters of hybridization buffer contains 875 μl of nuclease-free $H_2O$, 1.25 ml in situ hybridization salts, 5 ml of deionized formamide, 2.5 ml of 50% (wt/vol) dextran sulfate, 250 μl of 50× Denhardt's solution, and 125 μl of 100 mg/ml tRNA. Hybridized slides were then washed twice using 0.2× SSC buffer (saline-sodium citrate). To immobilize the hybridized probes, samples were incubated for 10 min in freshly prepared *N*-(3-Dimethylaminopropyl)-*N*'-ethylcarbodiimide hydrochloride (EDC) solution containing 0.13 M 1-methylimidazole, 300 mM NaCl (pH 8.0). Then, samples were incubated for 1 h and 15 min in 0.16 M EDC (item 03450, Sigma-Aldrich, St. Louis, MO) solution. Afterward, samples were neutralized in 0.5% (w/v) glycine and blocked in 1× blocking buffer (1% blocking reagent in 1× Tris Buffered Saline (TBS) buffer), and 1× washing buffer (1% wt/vol bovine serum albumin; Sigma-Aldrich-Aldrich, A7906) and 0.3% Triton x-100 in 1× TBS buffer) for 1 h each. Samples were then incubated with primary antibody overnight at 4 °C followed by 4× washes in 1× washing buffer, 15 min each. Samples were then incubated with a secondary antibody overnight at 4 °C followed by 4× washes in 1× washing buffer, 15 min each. After final wash in 1× TBS buffer, samples were mounted using ProLong™ Glass Antifade Mountant (ThermoFisher Scientific, P36980). Antibodies were also hybridized to non-labeled samples as negative controls. Probes are shown in Supplementary Data 12.

**smFISH for phasiRNAs**. Fifty probes were designed, corresponding to the top 50 21 nt phasiRNAs based on abundance in our libraries (i.e., the most abundant were selected). Each probe is 17–22 nt in length. The probes are designed using a web program: Stellaris Probe Designer (https://www.biosearchtech.com/support/tools/design-software/stellaris-probe-designer). The probes were ordered from LGC Biosearch Technologies with a 3'-end amino group and coupled with the fluorophore tetramethylrhodamine (TMR) manually[50]. Fresh, unopened strawberry buds were prepared, embedded, and sectioned the same way as for sRNA-FISH. Sample slides were then hybridized with smFISH probes at a concentration of 5 ng/μl in a 37 °C hybridization oven overnight. After hybridization, the slides were washed 3× with 100 ml washing buffer, 20 min each wash. Samples were then washed with 2× SSC buffer and were equilibrated for 2 min. Samples were mounted using ProLong™ Glass Antifade Mountant (ThermoFisher Scientific, P36980). smFISH probes for human AR mRNA were used as negative control. Probes are shown in Supplementary Data 12.

**Image acquisition**. Spectral imaging was conducted on a Carl Zeiss LSM 880 laser scanning microscopy capable of both LSCM and multiphoton microscopy. The Zen software (v2.3; Carl Zeiss) was used for both acquisition of spectral images and linear spectral unmixing. Spectral data for pure Alexa Fluor® fluorophores were used as positive controls and non-labeled samples were used to obtain autofluorescence spectra for linear spectral unmixing. Brightness and contrast of images in the same figure panel were adjusted equally and linearly in Zen software (Carl Zeiss).

**NanoPARE library construction, sequencing, and data analysis**. NanoPARE libraries were constructed from 5 ng total RNA obtained from pre-meiotic stage anthers of wild strawberry and columbine, following the published protocol[14] with these minor modifications: (i) 12 cycles of PCR was performed at the pre-amplifcation step; (ii) 1.5 μl tagmentation enzyme (TDE1, Illumina) was used; (iii) 12 cycles of PCR was performed for the final enrichment step; and (iv) 0.7 volume of AMPure XP beads (42 μl) was used for cleaning up each nanoPARE library.

These libraries were sequenced on a NextSeq 550 instrument to produce 76 nt single-end reads using custom primers as described[14]. NanoPARE libraries were trimmed of the adapter (5'-CTGTCTCTTATACACATCT-3') using a standard pipeline[29] and all reads were chopped to a uniform length of 20 nt, matching standard PARE data. Finally, sPARTA was used for the trigger-target validation using default parameters for PARE data[33].

## Data availability

Data supporting the findings of this work are available within the paper and its Supplementary Information files. A Reporting Summary for this Article is available as a Supplementary Information file. All the sRNA, RNA-seq, and nanoPARE data generated during this study were deposited in NCBI's SRA (Sequence Read Archive) under the accession number PRJNA669702. All the RNA-seq and sRNA data from public databases used for the analysis are listed in Supplementary Data 13. Source data are provided with this paper.

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

## Acknowledgements

We thank members of the Meyers lab for helpful discussions and Joanna Friesner for assistance with editing. We thank Mayumi Nakano for assistance with data handling. We also thank Michael Nodine and Michael Schon for providing advice for the nanoPARE experiments. This work was supported by resources from the Donald Danforth Plant Science Center and the University of Missouri—Columbia and by NSF IOS award 1754097. Microscopy equipment was acquired with a shared instrumentation grant (S10 OD016361) and access was supported by the NIH-NIGMS (P20 GM103446), the NSF (IIA-1301765), and the State of Delaware.

## Author contributions

B.C.M. and S.P. designed the research. S.P. performed the experiments and analyzed the data. K.H. performed fluorescent in situ hybridizations and imaging analysis. S.B. helped with staging of buds/anthers. J.Z. generated the nanoPARE data. J.L.C. supervised in situ hybridizations. E.M.K. collected buds and anthers of columbine. S.P. and B.C.M. wrote and revised the manuscript with input from all co-authors.

## Competing interests

The authors declare no competing interests.
