## [Peer Review File · Nature Communications]

REVIEWER COMMENTS

Reviewer #1 (Remarks to the Author):

This manuscript reports the analysis of sRNA-seq data and reveals the existence of 21-nt phased siRNAs (phasiRNA) in four eudicot species (wild strawberry, rose, columbine, flax), while previous studies reported that these small RNAs are absent from other well-studied eudicots. The analysis show that their putative miRNA triggers differ across species. In situ hybridization experiments in strawberry document the tissue localization of the putatively novel trigger miR11308 and its resulting phasiRNAs, although it is unclear what the comparisons reported reveal.

21-nt phasiRNAs are a specific class of small regulatory RNAs, whose role remains largely elusive (they have been involved in specific instances of male sterility in rice, but they still lack obvious targets and their physiological importance and the mechanisms by which they achieve their function remain largely unknown). The work presented is well executed (but see below for some caveats) and provides some welcome details on the phylogenetic distribution of these genetic elements. As such, it is clearly worth reporting. However, the insight we get is actually pretty limited for a general audience, both in terms of their evolutionary patterns and in terms of their functional role. First, as I understand these sRNAs were already reported in the outgroup *Picea*, so it was already clear that these regulatory RNAs were ancestrally present and were lost secondarily. In this context, the evolutionary insight is limited. Obtaining a proper phylogenetic distribution of which species have retained these sRNAs would have required the analysis of a much larger number of species. At present, it is hard to draw any general conclusion of how often/ under what circumstances these sRNAs have been lost. Second, the insight we get on their biological function also remains limited, and the functional analyses reported are limited to indirect correlations : e.g. the presence of a specific RDR6/AGO/DCL gene in the genome, which in itself does not provide proof that any of them is involved; e.g. their tissue localization which is a very indirect indication of what they actually do ; e.g. the identification of their predicted triggers, which are derived solely from bioinformatic predictions but with no direct proof.

Overall, I think that this work will be of great interest to the community of people studying the diversity of small RNAs, but is currently too limited to have a broad appeal to the larger community interested in the evolution of gene regulatory mechanisms.

Reviewer #2 (Remarks to the Author):

Recently, phasiRNAs have been increasingly recognized as an important class of regulatory RNAs in seed plants. The absent report of 21-nt phasiRNAs in eudicots, make our own prior assumption that they are absent from the eudicots. Here, the authors described the 21-nt phasiRNAs in eudicot plants triggered by miR2118 or other miRNAs. This discovery refreshed our knowledge that 21-nt phasiRNAs emerged both in eudicots and monocots. The writing is easy to read. My comments below provide potential

improvements for consideration.

1. All of the 21-nt PHAS loci in this manuscript was deduced by the bioinformatic tools. Could the author provide some other evidences using the experiment such as RACE or the Degradome datasets to validate the cleavage site of 21-nt PHAS regulated by miR2118 or other miRNAs?
2. In monocots, hundreds or thousands of 21-nt reproductive PHAS loci had been identified, while only dozens or more than one hundred of 21-nt PHAS loci were identified in this manuscript. Is th the difference between monocots and eudicots? Or the number of 21-nt PHAS loci is associated with the genome size?
3. In wild strawberry, the productions of 21-nt phasiRNAs were initiated by both miR11308 and miR2118 in the reproductive tissues. Is there difference in the abundance of 21-nt phasiRNAs triggered by miR11308 or miR2118?
4. In some eudicots, 21-nt reproductive phasiRNAs have not been reported in several well-studied plant families. Can we deduce that 21-nt reproductive phasiRNAs may not be very important, and can be gradually deleted in the future evolution process?
5. Is there miR11308 or aco-cand81 loci also in the monocot genomes?
6. Is there difference in the expression level between the U-21-nt phasiRNAs or C-21-nt phasiRNAs in eudicots?

Response to Reviewers

Reviewer #1:

This manuscript reports the analysis of sRNA-seq data and reveals the existence of 21-nt phased siRNAs (phasiRNA) in four eudicot species (wild strawberry, rose, columbine, flax), while previous studies reported that these small RNAs are absent from other well-studied eudicots. The analysis shows that their putative miRNA triggers differ across species. In situ hybridization experiments in strawberry document the tissue localization of the putatively novel trigger miR11308 and its resulting phasiRNAs, although it is unclear what the comparisons reported reveal.

21-nt phasiRNAs are a specific class of small regulatory RNAs, whose role remains largely elusive (they have been involved in specific instances of male sterility in rice, but they still lack obvious targets and their physiological importance and the mechanisms by which they achieve their function remain largely unknown). The work presented is well executed (but see below for some caveats) and provides some welcome details on the phylogenetic distribution of these genetic elements. As such, it is clearly worth reporting. However, the insight we get is actually pretty limited for a general audience, both in terms of their evolutionary patterns and in terms of their functional role. First, as I understand these sRNAs were already reported in the outgroup *Picea*, so it was already clear that these regulatory RNAs were ancestrally present and were lost secondarily. In this context, the evolutionary insight is limited. Obtaining a proper phylogenetic distribution of which species have retained these sRNAs would have required the analysis of a much larger number of species. At present, it is hard to draw any general conclusion of how often/under what circumstances these sRNAs have been lost. Second, the insight we get on their biological function also remains limited, and the functional analyses reported are limited to indirect correlations: e.g. the presence of a specific RDR6/AGO/DCL gene in the genome, which in itself does not provide proof that any of them is involved; e.g. their tissue localization which is a very indirect indication of what they actually do; e.g. the identification of their predicted triggers, which are derived solely from bioinformatic predictions but with no direct proof.

Overall, I think that this work will be of great interest to the community of people studying the diversity of small RNAs, but is currently too limited to have a broad appeal to the larger community interested in the evolution of gene regulatory mechanisms.

Response: Thank you for the comments. We have a few responses to specifically address some of your points. In case of the in situ hybridization experiments, the contribution is documenting the localization of a new trigger for the pre-meiotic phasiRNAs (miR11308 –the first trigger for premeiotic phasiRNAs that has been described that is not miR2118). We also show this for the resulting 21-nt phasiRNAs. We use the in situ data for an extremely precise, spatial-temporal analysis, which is not possible with small RNA sequencing. The result is that the abundance of the trigger is higher at pre-meiotic/meiotic stages while the 21-nt phasiRNAs are more abundant only at the pre-meiotic stages. Thus, the in situ experiments in different stages of anther development validate and enrich the observations from the sequencing experiments.

In terms of the broader appeal, we would point out that reproductive small RNAs have been the focus of innumerable studies in animals for ~15 years, while the work in plants is not as well advanced. We envision the focus of our work, pre-meiotic phasiRNAs, as analogous to mammalian pre-pachytene piRNAs (see, for example <https://www.nature.com/articles/cr201441>). Prior to our work, reproductive 21-nt phasiRNAs were reported in only a subset of monocots,

mainly in grasses, and assumed absent in eudicots, which comprise 75% of angiosperm species. While we agree that the phylogenetic work was limited, we did not claim it was extensive or exhaustive – we merely sought to show that there are diverse eudicot species that have this pathway. Our prior report of reproductive 21-nt phasiRNAs in male cones of *Picea abies* was quite limited, and was essentially a single data point, buried in that paper. We feel that the current report substantially enhances the importance of reproductive phasiRNAs in angiosperm biology, as they were clearly present at the evolutionary emergence of flowering plants.

We were also able to describe the diversification of triggers of reproductive 21-nt phasiRNAs in eudicots as well as their tight spatio-temporal expression. In rice, this same class of phasiRNAs are implicated in photoperiod-sensitive male sterility, and underlie two of the loci used for ~20% of the hybrid rice produced in China (the two-line P/TGMS system). The discovery of this pathway in eudicots opens new possibilities for both basic and applied studies.

Regarding the biological function of the reproductive phasiRNAs, indeed, this is poorly characterized, although it seems unreasonable to ask for us to wrap up everything about these molecules in one paper. In the case of animal piRNAs, there are over 1500 papers in Pubmed and studies continue on their functions, biogenesis, regulation, expression, etc. We have experiments underway, of course, but those surely will be the subject of separate, standalone papers. Finally, in the case of triggers of the 21-*PHAS* target loci, we have now experimentally validated most of the target-trigger interactions using nanoPARE data as described in revised manuscript.

Reviewer #2:

Recently, phasiRNAs have been increasingly recognized as an important class of regulatory RNAs in seed plants. The absent report of 21-nt phasiRNAs in eudicots, make our own prior assumption that they are absent from the eudicots. Here, the authors described the 21-nt phasiRNAs in eudicot plants triggered by miR2118 or other miRNAs. This discovery refreshed our knowledge that 21-nt phasiRNAs emerged both in eudicots and monocots. The writing is easy to read. My comments below provide potential improvements for consideration.

Response: Thank you for the positive comments.

1. All of the 21-nt PHAS loci in this manuscript was deduced by the bioinformatic tools. Could the author provide some other evidences using the experiment such as RACE or the Degradome datasets to validate the cleavage site of 21-nt PHAS regulated by miR2118 or other miRNAs?

Response: Thank you for the suggestion. It took us several additional months but we have now generated degradome data using the recently developed technique “nanoPARE” (low input PARE libraries). With these data, we validated the trigger-target interactions that we described for triggers of 21-*PHAS* loci in both wild strawberry and *Aquilegia*. This is now described in the revised manuscript. We could validate the trigger-target interaction for both the canonical trigger miR2118 as well as the lineage-specific triggers miR11308 and “miRcand81” (now submitted to miRBase and renamed as miR14051 in this revised manuscript).

2. In monocots, hundreds or thousands of 21-nt reproductive PHAS loci had been identified, while only dozens or more than one hundred of 21-nt PHAS loci were identified in this manuscript. Is

the difference between monocots and eudicots? Or the number of 21-nt PHAS loci is associated with the genome size?

Response: We observed that the number of 21-nt *PHAS* loci is quite variable across different eudicots, as it is in monocots. The short answer is that there's no pattern, and we don't have enough data yet to know whether there is a connection between the number of 21-nt *PHAS* loci and genome size, although genomes like rice don't seem to support this hypothesis. Please see the table below.

Species	Genome size (Mb)	Number of reproductive 21- PHAS
Wild strawberry	240	25
Rose	503	9
Columbine	300	91
Flax	373	11
Maize	2300	463
Rice	430	1843
Wheat	17000	9073

3. In wild strawberry, the productions of 21-nt phasiRNAs were initiated by both miR11308 and miR2118 in the reproductive tissues. Is there difference in the abundance of 21-nt phasiRNAs triggered by miR11308 or miR2118?

Response: Only 11 out of 76 loci triggered by miR2118 are reproductive-enriched (five of them are from disease resistance genes, six are non-coding) as shown in Supplementary Fig. 2A, the lowermost 11 loci are reproductive-enriched loci while all of the (25) loci triggered by miR11308 are reproductive-enriched (Fig 1A). It appears from following figure that, overall, the abundance of reproductive-enriched 21-nt phasiRNAs triggered by miR11308 is more than those triggered by miR2118. The locus-specific phasiRNA abundance is shown in heatmaps in Fig 1A, and Supplementary Fig. 2A and Supplemental Table S1.

4. In some eudicots, 21-nt reproductive phasiRNAs have not been reported in several well-studied plant families. Can we deduce that 21-nt reproductive phasiRNAs may not be very important, and can be gradually deleted in the future evolution process?

Response: We hypothesize that 21-nt reproductive phasiRNAs are important for their role in male fertility, as demonstrated in rice, but their absence in other lineages could suggest that in the this pathway was not selectively advantageous in those lineages, or that its functions were replaced by other mechanisms. As for a general statement about its importance or a prediction of depletion – well, each plant lineage faces different conditions and life histories, and what is important in one may be unimportant in another, and vice versa.

5. Is there miR11308 or aco-cand81 loci also in the monocot genomes?

Response: Apparently not. We searched for their presence by scanning the genomes of representative species, as well as by doing similarity search of these miRNA against miRBase version 22, and based on this result, we confirm that these are lineage-specific miRNAs.

6. Is there difference in the expression level between the U-21-nt phasiRNAs or C-21-nt phasiRNAs in eudicots?

Response: We looked at the expression level of 21-nt reproductive phasiRNAs in Columbine based on the first nucleotide, and found that A-21-nt phasiRNAs have highest overall abundance compared to others as shown below. We find a greater number of A-21-nt phasiRNAs compared to others, as shown below. We analyzed 400 21-nt phasiRNAs from 91 reproductive 21-PHAS loci which were expressed more than 3 RPM in all the libraries.

First nucleotide composition

■ U ■ C ■ A ■ G

REVIEWERS' COMMENTS

Reviewer #2 (Remarks to the Author):

Now, I have a small question as below.

In Figure 6, the authors mentioned that the 21-nt phasiRNAs could silencing genes in cis and trans. However, there was no evidence to validate it. Recently, there were publications to report that 21-nt phasiRNAs could target the coding genes in rice. The authors had generated the degradome data using the recently developed technique “nanoPARE” to validate the target sites of microRNAs in the 21-PHAS loci. Using these degradome datasets, could the authors further investigate whether the 21-nt phasiRNAs cleave their targets in eudicots? Is the silence mechanism between 21-nt phasiRNAs and the protein coding genes conserved in eudicots and monocots?

Response to Reviewer

Reviewer #2:

Now, I have a small question as below.

In Figure 6, the authors mentioned that the 21-nt phasiRNAs could silencing genes in *cis* and *trans*. However, there was no evidence to validate it. Recently, there were publications to report that 21-nt phasiRNAs could target the coding genes in rice. The authors had generated the degradome data using the recently developed technique “nanoPARE” to validate the target sites of microRNAs in the 21-PHAS loci. Using these degradome datasets, could the authors further investigate whether the 21-nt phasiRNAs cleave their targets in eudicots? Is the silence mechanism between 21-nt phasiRNAs and the protein coding genes conserved in eudicots and monocots?

Response: Thank you for the suggestion. We predicted that the function of reproductive 21-nt phasiRNAs would be *cis* and *trans* targeting of genes; this was based on published studies in other species such as maize and rice. Since we generated only two datasets of degradome data for each species strawberry and columbine, it would be difficult to confidently validate targets of 21-nt phasiRNAs. As these phasiRNA are hundreds or thousands in number in each species, confidently validating targets needs more replication and datasets from different stages of anther development, so we choose to focus on this analysis in future experiments. To clarify, we added following sentence in discussion and cited appropriate studies to address this issue. “Based on the roles in other species family members of AGO5 might load 21-nt phasiRNAs that were predicted to act in both *cis* and *trans* to modulate transcript levels.” We also modified the legend in the Figure 6 to address this.